# Prognostic and Immunotherapeutic Roles of KRAS in Pan-Cancer

**DOI:** 10.3390/cells11091427

**Published:** 2022-04-22

**Authors:** Kaixin Yang, Chengyun Li, Yang Liu, Xueyan Gu, Longchang Jiang, Lei Shi

**Affiliations:** 1School of Public Health, Lanzhou University, Lanzhou 730000, China; yangkx21@lzu.edu.cn (K.Y.); lichengyun@lzu.edu.cn (C.L.); guxy@lzu.edu.cn (X.G.); 2Gansu Provincial People’s Hospital, Lanzhou 730000, China; liuy_1003@163.com; 3Department of Vascular Surgery, Shanghai East Hospital, Tongji University School of Medicine, Shanghai 200120, China; 4Research Center for Translational Medicine, Shanghai East Hospital, Tongji University School of Medicine, Shanghai 200120, China; 5Transcriptional Networks in Lung Cancer Group, Cancer Research UK Manchester Institute, University of Manchester, Alderley Park, Manchester SK10 4TG, UK

**Keywords:** KRAS, pan-cancer, prognostic value, immune infiltration

## Abstract

KRAS is one well-established tumor-driver gene associated with cancer initiation, development, and progression. Nonetheless, comparative studies of the relevance of KRAS across diverse tumors remain sparse. We explored the KRAS expression and prognostic values in diverse cancer types via multiple web-based bioinformatics tools, including cBioPortal, Oncomine, PrognoScan, Kaplan–Meier Plotter, etc. We found that KRAS is highly expressed in various malignancies compared to normal cohorts (BRCA, CHOL, ESCA, HNSC, LIHC, LUAD, LUSC, and STAD) and less expressed in COAD, KIRC, READ, and THCA than in normal samples. We observed the dysregulation of the DNA methylation of KRAS in cancers and discovered that numerous oncogenic and tumor-suppressive transcription factors bind the KRAS promoter region. Pan-cancer analysis also showed that a high level of KRAS is associated with poor outcomes. Additionally, KRAS is remarkably correlated with the level of immune cell infiltration and tumorigenic gene signatures. In conclusion, our findings reveal novel insights into KRAS expression and its biological functions in diverse cancer types, indicating that KRAS could serve as a prognostic biomarker and is associated with immune infiltrates.

## 1. Introduction

Cancer is a leading cause of mortality and a significant impediment to increasing life expectancy worldwide. It is a disease that results from the transformation of normal cells into tumor cells that proliferate rapidly and uncontrollably, and spread to adjacent organs or tissues [1]. According to the statistics from the World Health Organization (WHO) 2020, cancer is responsible for approximately 10 million deaths, with the number of cancer patients continuing to increase, resulting in tremendous losses for individuals and society [2]. Despite major advances in therapy, such as surgery, chemotherapy, radiation therapy, targeted therapy, and immunotherapy, prognosis survival remains poor due to a range of factors, such as drug resistance, side effects, and other potential issues [3,4]. Therefore, it is critical to develop novel therapeutic alternatives and biomarkers for early diagnosis and treatment.

Emerging studies have discovered that genes bearing somatic mutations or alternations are capable of promoting tumors via the adoption of the pioneered advance of next-generation sequencing [5]. These mutations, so-called “cancer drivers” because of their ability to contribute to tumorigenesis, offer selective advantages to cells in somatic tissue in comparison to neighboring cells, thereby affecting the tumor microenvironment and rewriting cellular metabolism [6]. Oncogenes and tumor suppressor genes are the two main types of genes that play roles in cancer. Oncogenes can activate the occurrence of the mutations by gene amplification or missense, whereas tumor suppressors silence or inactivate the mutated phenotypes via focal deletions or nonsense, frameshift, and splice-site across the gene transcripts [7,8].

The human proto-oncogene *RAS*, which contains three sub-categories (*HRAS*, *KRAS*, and NRAS), is one of the most frequently mutated genes in cancers, with gain-of-function mutations occurring in approximately 30% of human malignancies [9]. The *KRAS* mutation, a main carcinogenesis driver that has been extensively investigated in the last 30 years, is involved in ~90% of pancreatic cancer, 40% of colon cancer, and 20–30% of lung cancer [10]. *KRAS* can activate a plethora of signaling pathways by the direct engagement of effector molecules, the GTPase transductor proteins. KRAS cycles between guanosine diphosphate (GDP)-bound (inactive isoform) and guanosine triphosphate (GTP)-bound (active isoform) features. This conformational transition is modulated by guanine nucleotide-exchange factors (GEFs) that catalyze the exchange from GDP to GTP, or GTPase-activating proteins (GAPs), which stimulate GTP hydrolysis to GDP [11]. *KRAS* is one of the most important sensors that stimulates a wide range of effector molecules, including RAF, PI3K, and MAPK, allowing the transmission of transducing signals from the cell surface to the nucleus, as well as affecting cellular programming, such as cell proliferation, differentiation, and apoptosis that lead to tumorigenesis [12,13]. 

In this study, we used multiple public databases to analyze and summarize the mutation and expression of *KRAS* in different cancer types, including cBioPortal, Tumor Immune Estimation Resource (TIMER), The Cancer Genome Atlas (TCGA), UALCAN, Kaplan–Meier Plotter, etc. We first explored the mutation and expression of KRAS in different types of cancer. Following that, we investigated the DNA methylation level and the upstream regulators of *KRAS*, such as transcription factors and microRNAs, which could help us understand the aberrant level of *KRAS* in diverse cancers. We next analyzed the underlying function of *KRAS* in prognosis and the immune response. We also performed Gene Set Enrichment Analysis (GSEA), Gene Ontology, and Kyoto Encyclopedia of Genes and Genomes (KEGG) analysis to examine the enrichment of *KRAS* in biological activities and gene signatures. In summary, we conducted a thorough investigation to explore the prognostic values of *KRAS* as a clinical biomarker, thus providing useful in-depth information for tumor-targeting therapy.

## 2. Materials and Methods

### 2.1. cBioPortal

cBioPortal (https://www.cbioportal.org/, last accessed date 10 April 2022) is a user-friendly tool that integrates the data from the projects of TCGA and the International Cancer Genome Consortium (ICGC) that can visualize and analyze multidimensional cancer genomics data [14,15]. It is an open-source tool for interactively exploring multidimensional cancer genomics data sets, allowing users to visualize the gene expression, gene copy number alteration, gene mutation, overall survival, and clinical patient information without requiring bioinformatics expertise in 33 cancer types, including adrenocortical carcinoma (ACC), acute myeloid leukemia (AML), bladder urothelial carcinoma (BLCA), breast invasive carcinoma (BRCA), cervical squamous cell carcinoma (CESC), cholangiocarcinoma (CHOL), colon adenocarcinoma (COAD), lymphoid neoplasm diffuse large B cell lymphoma (DLBC), esophageal carcinoma (ESCA), glioblastoma (GBM), brain lower grade glioma (LGG), head and neck squamous cell carcinoma(HNSC), kidney chromophobe (KICH), kidney renal clear cell carcinoma (KIRC), kidney renal papillary cell carcinoma (KIRP), liver hepatocellular carcinoma (LIHC), lung adenocarcinoma (LUAD), lung squamous cell carcinoma (LUSC), mesothelioma (MESO), ovarian serous cystadenocarcinoma (OV), pancreatic adenocarcinoma (PAAD), pheochromocytoma and paraganglioma (PCPG), prostate adenocarcinoma (PRAD), rectum adenocarcinoma (READ), sarcoma (SARC), skin cutaneous melanoma (SKCM), stomach adenocarcinoma (STAD), testicular germ cell tumors (TGCT), thyroid carcinoma (THCA), thymoma (THYM), uterine corpus endometrial carcinoma (UCEC), uterine carcinosarcoma (UCS), and uveal melanoma (UVM).

### 2.2. Oncomine Database

The Oncomine Platform (https://www.oncomine.org, last accessed date 5 August 2021) is a comprehensive web-based tool with rigorous and peer-reviewed analytical methodologies and a powerful collection of analysis functions that compute gene expression signatures, clusters, and gene-set modules aimed at providing solutions for individual researchers and multinational organizations. It collects more than 700 independent datasets with high quality and has become a gold-standard tool for cancer research applications. We used the Oncomine database to explore *KRAS* expression in multiple cancer types with a threshold value of *p*-value < 0.05 and fold change > 1.5 [16].

### 2.3. TIMER

The Tumor Immune Estimation Resource (TIMER, https://cistrome.shinyapps.io/timer/, last accessed date 8 March 2022) is a comprehensive resource for assessing immune infiltration in 10,897 samples across 32 cancer types [17,18]. We analyzed the expression of *KRAS* in various cancer types and the correlation between KRAS and the expression of 6 immune infiltrates (B cells, CD4+T cells, CD8+T cells, neutrophils, macrophages, and dendritic cells) in multiple cancer types.

### 2.4. Survival Analysis in PrognoScan and Kaplan–Meier Plotter

We used the online tools PrognoScan (http://dna00.bio.kyutech.ac.jp/PrognoScan/, last accessed date 5 August 2021) and Kaplan–Meier Plotter (http://kmplot.com/analysis/, last accessed date 10 April 2022) to explore the correlation of the *KRAS* level with the patient survival time in various cancer types. PrognoScan is a novel database to determine the correlation between gene expression and prognostic values, such as overall survival (OS), disease-free survival (DFS), and distant metastasis-free survival (DMFS) across a large collection of publicly accessible cancer microarray datasets [19]. The threshold of a Cox *p*-value < 0.05 was applied in this analysis. The Kaplan–Meier Plotter with the source data derived from GEO, EGA, and TCGA is capable of accessing the effect of genes (mRNA, microRNA, and protein) on survival in 21 cancer types [20]. Herein, we used Kaplan–Meier Plotter to check the influence of *KRAS* on OS and relapse-free survival (RFS) across different types of cancers. Hazard ratio (HR) values with 95% confidence intervals and log-rank *p*-values were calculated.

### 2.5. Gene Signature Analysis

TCGA gene expression data were acquired from the web-based tool Broad GDAC Firehose and divided into two sub-categories (KRAS Low and KRAS High) based on the median level of KRAS. Next, we performed Gene Set Enrichment Analysis (GSEA) to determine the potential molecular mechanisms of KRAS in various cancer types [21]. The R package ggplot2 was used to generate a Bubble Chart to visualize the common gene signatures across LUAD, LUSC, BRCA, and PAAD.

### 2.6. SurvExpress

SurvExpress (http://bioinformatica.mty.itesm.mx:8080/Biomatec/SurvivaX.jsp, last accessed date 17 September 2021) is a web-based tool that collects over 20,000 samples from 130 datasets with censored clinical information encompassing more than 20 cancer types [22]. We investigated the prognostic values of KRAS in lung, ovarian, kidney, colon, and stomach cancer in the present study.

### 2.7. UALCAN

UALCAN (http://ualcan.path.uab.edu/index.html, last accessed date 10 April 2022) is a comprehensive, user-friendly, and interactive web resource to analyze publicly available cancer OMICS data (TCGA, MET500, and CPTAC) [23]. In this study, we analyzed the KRAS expression as well as the methylation level of KRAS promoters across diverse cancer types. The Beta value indicates the level of DNA methylation, ranging from 0 (unmethylated) to 1 (fully methylated). Different cut-offs of beta values have been proposed to indicate hypermethylation [Beta value: 0.7–0.5] or hypo-methylation [Beta-value: 0.3–0.25], respectively [24,25].

### 2.8. SurvivalMeth

SurvivalMeth (http://bio-bigdata.hrbmu.edu.cn/survivalmeth/, last accessed date 10 April 2022) was used to investigate the methylation level of KRAS promoters in LUAD, PAAD, BLCA, and BRCA [26].

### 2.9. ChIP-Atlas

ChIP-Atlas (https://chip-atlas.org/, last accessed date 20 April 2022) is an integrative and comprehensive database that allows users to study public epigenetic datasets, including ChIP-seq, DNase-seq, ATAC-set, and Bisulfite-seq [27]. Here, we used the IGV peak browser to visualize the binding regions of transcriptional factors (TFs) on KRAS-promoter regions derived from ChIP-Atlas data.

### 2.10. Protein–Protein Interaction (PPI)

ChIPBase v3.0 was used to explore the potential regulators of KRAS in diverse cancers [28]. Next, we employed an online database called String to examine the protein–protein interaction [29] and further used CytoScape, an open-source software, to display the PPI network [30].

### 2.11. TargetScan

TargetScan predicts the biological targets of microRNAs by searching for the presence of conserved 8mer, 7mer, and 6mer sites that match the seed region of each microRNA [31]. Here, we utilized CytoScape to visualize the network of possible microRNA targets for KRAS in this study.

### 2.12. Statistical Analysis

The statistic values were calculated using Student’s *t*-test. For survival analysis, the HR and *p*-value were calculated using univariate Cox regression analysis. A *p*-value < 0.05 was considered a significant threshold in this study.

## 3. Results

### 3.1. KRAS Mutation in Diverse Cancer Types

Gene mutations have been identified as the essential tumor drivers across various cancer types [6]. As a result, we are curious to know the most frequently mutated genes in various cancers. From the cBioPortal dataset, we discovered the top 20 most frequently mutated genes. Overall, TP53 is the most frequently mutated gene, and its mutations were found in 4796 of 12,538 samples, accounting for 38.25% of all cohorts (Figure 1A and Table 1). KRAS is also one of the most frequently mutated genes, accounting for 10.63% (1333 of 12,538) of all human tumors, particularly in PAAD, COAD, READ, LUAD, and UCEC (Figure 1B). We and other groups have previously established that the KRAS mutation drives lung cancer, pancreatic cancer, and colon cancer [32,33,34]. However, the studies of KRAS in pan-cancer are limited and we therefore focused on the expression and functions of KRAS in different cancer types. First, we checked the correlation between KRAS mutation and OS survival, and discovered that KRAS mutation does not affect survival outcomes (Figure 1C). Further correlation shows that the KRAS copy number is positively correlated with its mRNA level (Figure 1D). Additionally, KRAS is substantially expressed in gain and amplification cohorts compared to deep deletion, shallow deletion, and diploid samples (Figure 1E), suggesting that high copy numbers correlate with enforced KRAS levels.

### 3.2. KRAS Expression in Diverse Cancers

To gain a better knowledge of KRAS, we first examined KRAS expression in a variety of cancers. We analyzed the messenger RNA level of KRAS in different cancer types using the Oncomine database and found that KRAS is highly expressed in breast cancer, kidney cancer, lung cancer, myeloma, ovarian cancer, pancreatic cancer, and sarcoma compared to normal samples (Figure 2A). In addition, we used an open-source database TIMER to explore the expression of KRAS in the TCGA dataset and found a high expression of KRAS in BRCA, CHOL, ESCA, HNSC, LIHC, LUAD, LUSC, and STAD and a low level of KRAS in COAD, KIRC, READ, and THCA compared to normal samples (Figure 2B). Next, we used a different online source, UALCAN, to examine the expression of KRAS in different cancer types and found comparable results to those obtained from TIMER. KRAS is highly expressed in BRCA, CESC, CHOL, ESCA, HNSC, LIHC, LUAD, LUSC, PAAD, PCPG, STAD, and UCEC, whereas it has low expression in COAD, GBM, KIRC, READ, and THCA (Figure 2C). In addition, we employed a web-based tool named starBase, and found that KRAS is up-regulated in BRCA, CHOL, ESCA, HNSC, LIHC, LUAD, LUSC, STAD, and UCEC in tumor samples compared to adjacent normal tissues, which are consistent with findings from the TIMER and UALCAN datasets (Table 2).

### 3.3. DNA Methylation of KRAS

We next examined the potential reason why KRAS is aberrantly expressed in diverse cancer types. Accumulating studies have revealed that DNA methylation can regulate gene expression by modulating transcription. Aberrant DNA methylation, including hypermethylation and hypomethylation, promotes malignant tumorigenesis. DNA hypermethylation stimulates methylation and results in transcriptional suppression and gene silencing; in contrast, DNA hypomethylation indicates the defect of DNA methylation and improves aneuploidy [35]. In this study, we used the online tool UALCAN to investigate the DNA methylation of KRAS and its related prognostic values in different cancer types. We observed a significantly increased hypomethylation level of the KRAS promoter in LUAD, KIRC, KIRP, PAAD, CHOL, CESC, and HNSC, which is consistent with the enforced level of KRAS in tumors compared to normal tissue (Figure 3A–D, Appendix A). The hypomethylation of KRAS in BLCA, READ, and COAD is low compared to their normal counterparts, which matched the low level of KRAS in these tumor cohorts (Figure 3F,G, Appendix A). Interestingly, the hypomethylation of KRAS is decreased in LUSC, BRCA, ESCA, and LIHC, even though we found a significant upregulation of KRAS in these tumors (Figure 3H,I, Appendix A), suggesting a complex tumor microenvironment, and further mechanisms in this manner are needed.

Next, we used SurvivalMeth to investigate the correlation between the DNA methylation of KRAS and prognostic values in different cancer types. As shown in Figure 4, different sites of the DNA methylation of KRAS are less expressed in tumors than in normal samples in the LUAD and PAAD, and increased in READ and COAD (Figure 4A–D, Table 3), which is consistent with the mRNA expression level and therefore implies the potential important connection between DNA methylation and gene expression.

### 3.4. Regulators of KRAS

Recent research has found that transcription factors modulate KRAS-dependent carcinogenesis [36,37]. We next used ChIPBase 3.0 to search for the potential transcription factors that may activate or repress the expression of KRAS. A total of 34 transcriptional factors were identified, which bind to 1Kb upstream of the KRAS transcriptional start site (TSS) (Appendix A). Next, we explored ChIP data via ChIP-Atlas to determine if these 34 transcriptional factors bind to the KRAS promoters. IGV Gene Brower carrying the binding signals shows that these genes transcriptionally bind to the KRAS promoter (Figure 5A). Additionally, a PPI network with CytoScape elucidated the interaction among the 34 transcriptional factors and KRAS, suggesting that KRAS plays a wide range of roles through the complex networks (Figure 5B).

In the past two decades, microRNAs have been identified as versatile gene regulators [38]. MicroRNAs interact with the 3’untraltated region (3’UTR) of target mRNAs to induce mRNA degradation and translational repression. We performed in silico analysis through TargetScan and identified 52 microRNAs that target KRAS. (Appendix A). The PPI network depicts the interaction between KRAS and microRNAs (Figure 5C). Considering the critical roles of these microRNAs in cancer, such as miR-7-5p and miR-181-5p [39,40], we speculate that KRAS may participate in these tumor-relevant mechanisms through microRNA.

### 3.5. Prognostic Value of KRAS in Various Cancers

Next, we examined the prognostic value of KRAS in various cancers. We established a substantial association between the KRAS level and OS in six types of cancer using PrognoScan, which collects data mainly from the Gene Expression Omnibus database. High expression of KRAS is associated with poor prognosis in blood cancer (GSE2658, HR = 0.55, Cox *p*-value = 0.00147, Figure 6A), breast cancer (GSE11121, HR = 1.45, Cox *p*-value = 0.000745. Figure 6B; GSE2034, HR = 1.18, Cox *p*-value = 0.000017, Figure 6C), colorectal cancer (GSE17537, HR = 1.83, Cox *p*-value = 0.000009, Figure 6D; GSE17537, HR = 0.97, Cox *p*-value = 0.008760, Figure 6E), lung cancer (Jacob-00182-UM, HR = 0.60, Cox *p*-value = 0.014977, Figure 6F), ovarian cancer (GSE9891, HR = 0.36, Cox *p*-value = 0.009022, Figure 6G), and soft tissue cancer (GSE30929, HR = 1.01, Cox *p*-value = 0.000002, Figure 6H).

We then examined the association between the KRAS level and the prognosis of cohorts with Kaplan–Meier Plotter. The survival metrics include OS and RFS. The Kaplan–Meier survival curves indicate that KRAS is associated with poor OS in BRCA (Figure 7A), CESC (Figure 7B), ESCA (Figure 7C), LUAD (Figure 7D), PAAD (Figure 7E), LIHC (Figure 7F) whereas with protective OS in KIRP (Figure 7G), LUSC (Figure 7H), KIRC (Figure 7I) and READ (Figure 7J). We also examined the relationship between KRAS and DFS in cancer patients. KRAS expression has a deleterious effect on RFS in six types of cancer, including BRCA (Figure 7A), CESC (Figure 7B), LUAD (Figure 7D), PAAD (Figure 7E), KIRP (Figure 4G), and LUSC (Figure 7H), and protectively affects RFS survival in KIRC (Figure 7I). In addition, univariate analysis with the Cox proportional hazard model shows that KRAS expression is significantly correlated with OS in lung cancer in different datasets (Appendix A), indicating the important potential of KRAS in these cancer types.

We were also curious if KRAS correlates with distinct clinicopathological features; therefore, we selected BRCA and LUAD as examples, because KRAS is highly expressed in these two cancer types. KRAS is significantly associated with BRCA OS of patients at stages 2, 3, and 4, as well as LUAD OS at stages 1 and 2 (Table 4). KRAS is remarkably more correlated with BRCA and LUAD OS in samples harboring KRAS mutations than non-mutations (HR 1.9 vs. 1.54 in BRCA; HR 1.72 vs. 1.54 in LUAD) (Table 4). Interestingly, KRAS mainly affects BRCA OS in female patients. However, it has a greater impact on males than females in LUAD, which may be because of the higher smoking record of men than women. 

Furthermore, we used an online biomarker validation tool called SurExpress to determine the gene level and patient survival outcomes in cancers. We observed that the high-risk group had a higher KRAS mRNA level than the low-risk group, and the high-risk group presented worse OS outcomes in lung cancer, ovarian cancer, kidney cancer, colon cancer, and stomach cancer (Figure 8, Appendix A). Moreover, the abnormal level of KRAS showed substantial correlations with drug–gene interactions in different cancer types (Table 5).

### 3.6. Immune Cell Infiltration of KRAS in the Pan-Cancer Analysis

Inflammation is a common hallmark of cancer and substantially plays a crucial role in tumor development and progression [41,42]. It has been well-established that KRAS influences tumor inflammation and the immune response, and consequently affects tumor initiation, invasion, and metastasis [43,44,45,46]. We therefore examined the coefficients of correlations between the KRAS level and immune infiltration in various cancer types via the TIMER database. The results reveal that the expression levels of KRAS are significantly correlated with the infiltration levels of B cells, CD8+T cells, CD4+T cells, macrophages, neutrophils, and dendritic cells in 10, 13, 8, 9, 11, and 7 cancer types, respectively (correlation threshold ∣ R > 0.2, *p* < 0.05) (Figure 9 and Appendix A). Among all of them, KRAS is positively correlated with CD8+ T cells, macrophages, and neutrophils in BRCA (Figure 9A); positively with CD8+T cells in ESCA (Figure 9B); positively correlated with CD4+T cells in HNSC (Figure 9C); positively correlated with B cells, CD8+T cells, CD4+T cells, macrophages, neutrophils, and dendritic cells in LIHC (Figure 9C); positively correlated with CD8+T cells and neutrophils in UCEC (Figure 9H); positively correlated with B cells, CD8+T cells, and neutrophils in BLCA; positively correlated with B cells, CD8+T cells, macrophages, and neutrophils in CHOL; positively correlated with B cells, CD8+T cells, CD4+T cells, macrophages, neutrophils, and dendritic cells in COAD; and positively correlated with B cells, CD8+T cells, macrophages, and dendritic cells in KICH (Appendix A). However, we observed a negative correlation between KRAS and CD4+T cells in UCEC (Figure 9H), and between KRAS and CD4+T cells in KICH (Appendix A). All these data suggest that KRAS affects tumorigenesis and the prognostic survival of patients via immune infiltration. 

### 3.7. Gene Signature Analysis in LUAD, LUSC, BRCA, and PAAD TCGA Datasets

Following that, we inquired about the *KRAS*-relevant gene signatures in various malignancies. We downloaded the TCGA datasets of LUAD, LUSC, BRCA, and PAAD due to the well-established oncogenic roles of KRAS in these tumors compared to normal samples and acts as a deleterious prognostic factor for the association with the poor outcome (Figure 2A–C, Figure 6B,C,F, Figure 7A,D,E,H, Figure 8). We first divided each dataset into two sub-categories (KRAS Low and KRAS High) in accordance with the median expression of KRAS. Gene Set Enrichment Analysis for Hallmark gene sets, C2 curated gene sets, C5 Ontology gene sets, C6 Oncogenic signature gent sets, and C7 Immunologic signature gene sets was applied to explore the putative molecular mechanisms. Intriguingly, important tumor-related signatures, such as “G2M checkpoint”, “EMT”, and “apoptosis” were commonly enriched in four different datasets, indicating the important roles of KRAS in cancers (Figure 10A–D). Additionally, Gene Ontology analysis also shows the unique functions of KRAS in biological processes (BPs), cellular components (CCs), and molecular functions (MFs). Specifically, we observed the enrichment of RNA destabilization and chemokine activity in LUAD, RNA splicing, mRNA processing, and immune receptor activity in LUSC (Figure 10E), DNA repair, DNA damage, and DNA binding in BRCA (Appendix A), and cytokinesis, ribosome, and cadherin binding in PAAD (Appendix A). We also observed the enrichment of several immunological signatures, such as “TNFα/NFκB signaling” and “immune cells”, which was consistent with the immune infiltration findings, suggesting that KRAS may affect the anti-tumor immune response (Figure 10A,D and Figure 11). All of the common and exclusive signatures imply the versatile functions of KRAS in tumors.

## 4. Discussion

Increasing studies have identified that KRAS is one of the most frequently mutated genes that impacts a multidomain group of intracellular signaling pathways that are involved in tumor cell growth, survival, and metastasis. KRAS mutation is a hallmark of cancer and hampers the association of GTPase-activating proteins, which leads to GTP hydrolysis, therefore stabilizing effector binding and stimulating KRAS signaling [47]. There are more than 10 different types of KRAS mutations, including KRAS G12D, G12V, G12C, G13D, Q61H, G12R, G12A, G12S, A146T, and G61L [48]. The TCGA database indicates that G12D (17.31%) is the most frequently mutated type, followed by G12V (13.66%) and G12C (6.46%) in various cancer types. KRAS mutations, which are one of the most promising and yet unresolved clinical observations in KRAS onco-biology, vary between human cancers in line with the positions and types of substitution of the mutations. Regarding the variability in the mutated position, a well-established example is that approximately 90% of the KRAS mutations occur at codon 12 in cancer. In addition, the G12C mutation more frequently occurs in non-small-cell lung cancer (NSCLC), while G12D is predominant in PDAC, COAD, and AML [49,50]. The active type of KRAS enhances the interaction with upstream effectors and consequently triggers multiple signal-transduction pathways, including the RAF-MEK-ERK (MAPK) and phosphatidylinositol 3-kinase (PI3K) pathways [51]. 

In this meta-analysis, we discussed recent advances in the research of KRAS in cancer, with an emphasis on the KRAS expression levels and their biological functions in various cancer types, which provides better knowledge for the exploration of prognosis biomarkers and immunotherapeutic targets in clinical strategies (Figure 12). We observed that KRAS mutations frequently occur in PAAD, COAD, READ, LUAD, etc. Moreover, KRAS mutations are significantly correlated with the KRAS levels in tumor cohorts, suggesting that KRAS expression is associated with the oncogenic roles. We next explored KRAS expression in different cancers compared to normal tissue, and discovered that it is highly expressed in eight cancers (BRCA, CHOL, ESCA, HNSC, LIHC, LUAD, LUSC, and STAD) while it has low expression in four cancers (COAD, KIRC, READ and THCA).

DNA methylation represents a key epigenetic modification that regulates chromatin architecture [52]. In this manuscript, we conducted a wide investigation by visualizing the DNA methylation of KRAS in pan-cancer patterns to understand how this important epigenetic mark alters KRAS expression during tumorigenesis. We observed high hypomethylation in LUAD, KIRC, KIRP, PAAD, CHOL, CESC, and HNSC, and low hypomethylation in BLCA, READ, and COAD compared to their normal counterparts. In line with the KRAS expression in cancers, these results suggest that DNA methylation particulates in the KRAS function in cancers. In addition, recent findings from large-scale analyses of gene expression profiles indicate that DNA methylation could affect the gene levels via its interaction with transcription factors [53]. Therefore, we asked if transcriptional factors are responsible for DNA methylation and gene alteration of KRAS. By utilizing in silico analysis, we discovered 34 transcription factors with putative binding regions on 1kb upstream or downstream of the KRAS transcription start sites. Importantly, all of these transcription factors are engaged in multiple cell processes throughout cancer development. For example, E2F1 could facilitate breast cancer carcinogenesis by promoting cirSEPT9 biogenesis [54]. ZEB1, a well-known regulator of EMT, drives migration and metastasis, and appears to be a central switch in cancer cell determination [55]. The PPI network reveals the interaction among these transcription factors and KRAS, indicating the essential functions of KRAS in oncogenic signaling.

Following that, we looked at the prognostic values of KRAS in different cancer types. The findings conclude that high expression of KRAS is significantly correlated with poor outcomes in BRCA, CESC, ESCA, LUAD, PAAD, and LIHC, whereas it is relevant to protective prognosis in KIRP, KIRC, and READ. Intriguingly, these prognostic values match their expression in diverse cancers. Though a high level of KRAS in LUSC implies a beneficial prognosis (overall survival), it still shows a poor correlation with RFS in LUSC (Figure 7H). The analysis of the Cox proportional hazards model from multiple databases illustrated that a high KRAS level is associated with lung cancer (Appendix A). Moreover, we observed a paradox that KRAS has low expression in COAD, but a high level of KRAS is still relevant with poor overall in colon cancer. Chun et al. previously validated that KRAS presents an oncogenic role by modulating mitochondrial metabolism in colon cancer by inducing HIF-1α/HIF-2α target genes [56]. Considering the high rate (~30%) of KRAS mutations in colon cancer [49], we thereby hypothesize that the functions of KRAS may vary in different tumor microenvironments, and further investigation is urgently needed.

Immune cells infiltrating tumors engage in broad and dynamic crosstalk with cancer cells [57]. One of the major studies has stated that those genetic alterations of oncogenes or tumor suppressors encoding inflammasome components frequently confer susceptibility to tumorigenesis [58,59]. Given the essential roles of KRAS in diverse cancers, we further systematically analyzed the correlation between KRAS and tumor immunity. We found that KRAS is substantially correlated with tumor immune cell infiltration, including B cells, CD8+T cells, CD4+T cells, macrophages, neutrophils, and dendritic cells, suggesting that KRAS may affect tumor processing and prognosis through cancer immunity. To better understand the KRAS-related functions in tumorigenesis and cancer immunity, we further explored the molecular characteristics of genes by utilizing gene set enrichment analysis in four types of cancers (LUAD, LUSC, BRCA, and PAAD). We split the data into two groups according to the median level of KRAS and observed that some important signatures (EMT, apoptosis, and TNFα/NFκB signaling) and immune cell infiltration were inclusively enriched in the KRAS-high vs KRAS-low datasets in these cancers. EMT is a well-known process playing a critical role in tumor migration, metastasis, and chemotherapy [60]. TNFα is a major inflammatory cytokine in maintaining the immune system, inflammation. and host defense systems [61]. NFκB refers to a family of transcription factors that regulate a wide spectrum of biological processes, including inflammation, proliferation, and tumor development [62]. These findings strongly support our results and show the immunological roles of KRAS in diverse tumor types.

Oncogenic KRAS mutations lead to cancer chemoresistance targeting receptor target kinases (RTKs) [63]. Extensive efforts to silence KRAS in the past three decades have proven fruitless, and it has been considered an undruggable target of cancer. Recently, specific inhibitors (AMG510 and MRTX849) against G12C have offered an effective strategy to treat KRAS-driven cancers, suggesting that a range of mutant KRAS allele-specific targeted compounds could be druggable [64,65]. However, there will be a continued challenge to developing specific inhibitors for other variants, such as KRAS G12D and G12V, which are the most common KRAS variants that are associated with the largest patient populations.

There are some drawbacks to the present study. First, conflicts may exist, since the data were derived from multiple bioinformatics resources. In addition, the present study must be further validated in vitro and in vivo to determine the precise mechanisms of KRAS in diverse cancer types.

In summary, this article outlines the fundamental roles of KRAS that govern the effects of tumor inflammation and provides insights into the development of novel prognostic biomarkers for early diagnosis and potential cancer therapeutics.

## 5. Conclusions

Pan-cancer analysis of KRAS indicated that 33 cancers had different expressions of these genes between normal and tumor samples.KRAS could serve as a key prognostic factor in different cancer types.KRAS could affect tumor development through tumor immune cell infiltration.Our study illustrates the characterization of KRAS expression in various cancer types and highlights its potential value as a predictive biomarker, which sheds light on the further investigation of the prognostic and therapeutic potential of KRAS inflammation.

## Figures and Tables

**Figure 1 cells-11-01427-f001:**
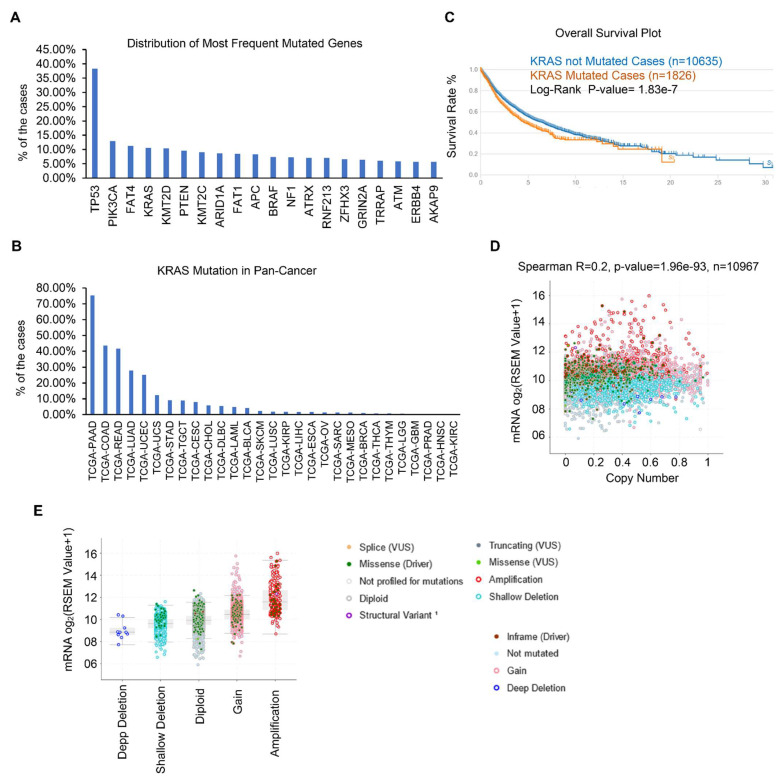
KRAS is frequently mutated in different cancer types. (**A**) Top 20 frequently mutated genes in different cancer types. (**B**) Frequency of KRAS mutations in various cancer types. (**C**) Survival blot showing the correlation between clinical outcomes and KRAS mutations. (**D**) Correlation between KRAS copy number and KRAS mRNA level in TCGA cohorts. (**E**) KRAS is highly expressed in KRAS amplification and gain cohorts than in deep deletion and shallow deletion tumor samples.

**Figure 2 cells-11-01427-f002:**
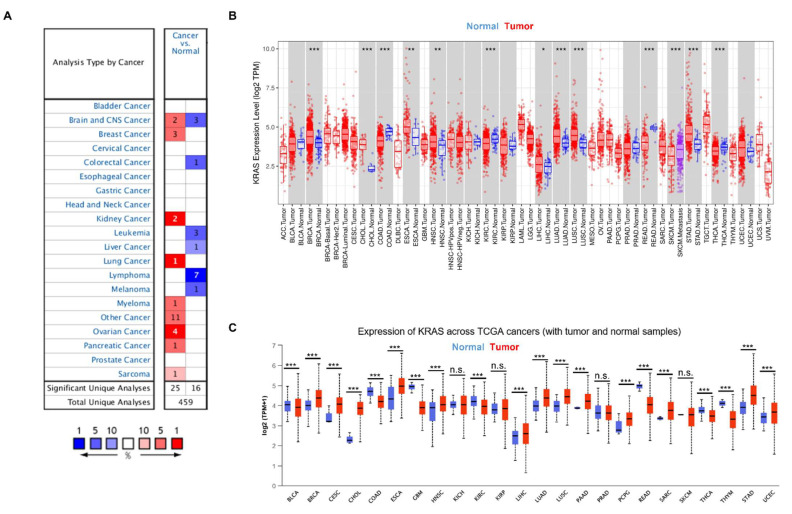
KRAS expression in multiple cancers. (**A**) High or low expression of KRAS in different cancer tissues compared with normal tissues from the Oncomine dataset. The numbers in each cell indicate the number of datasets in each cancer type. (**B**,**C**) Human KRAS expression levels in different types from TCGA data in TIMER (**B**) and UALCAN (**C**). *p*-value * < 0.05, *p*-value ** < 0.01, *p*-value *** < 0.001.

**Figure 3 cells-11-01427-f003:**
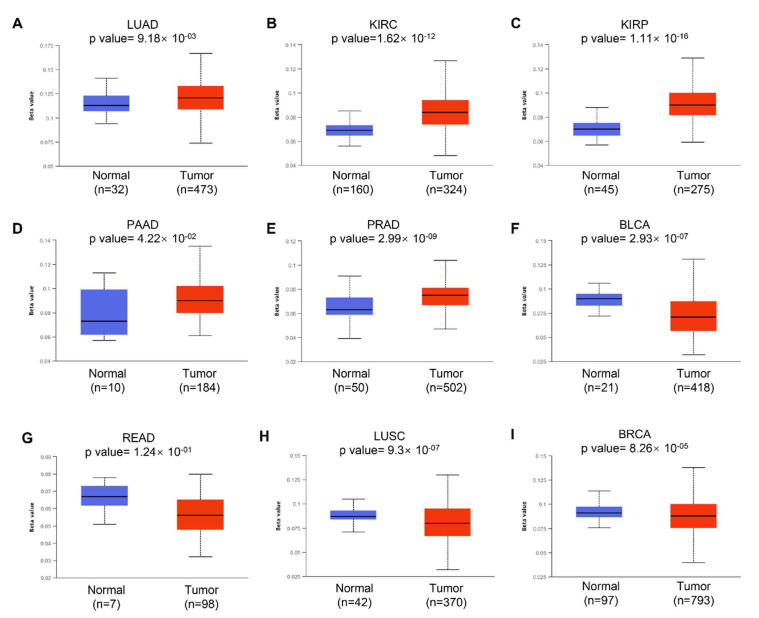
Histogram bars showing the hyper- or hypomethylation of the KRAS promoter region in tumors compared to normal tissues in LUAD (**A**), KIRC (**B**), KIRP (**C**), PAAD (**D**), PRAD (**E**), BLCA (**F**), READ (**G**), LUSC (**H**), and BRCA (**I**). A *p*-value < 0.05 was considered a significant threshold.

**Figure 4 cells-11-01427-f004:**
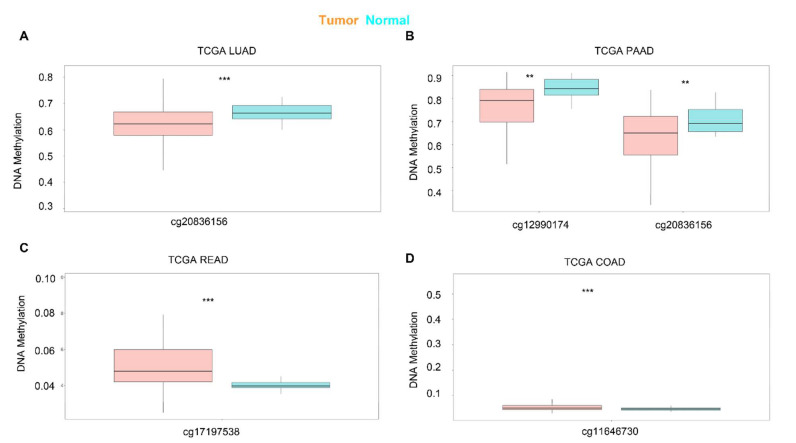
Expression level of the DNA methylation of KRAS in cancers. DNA methylation of KRAS has a low level of expression in LUAD (**A**) and PAAD (**B**), and a high level of expression in READ (**C**), and COAD (**D**) compared to normal samples. *p*-value ** < 0.01, *p*-value *** < 0.001.

**Figure 5 cells-11-01427-f005:**
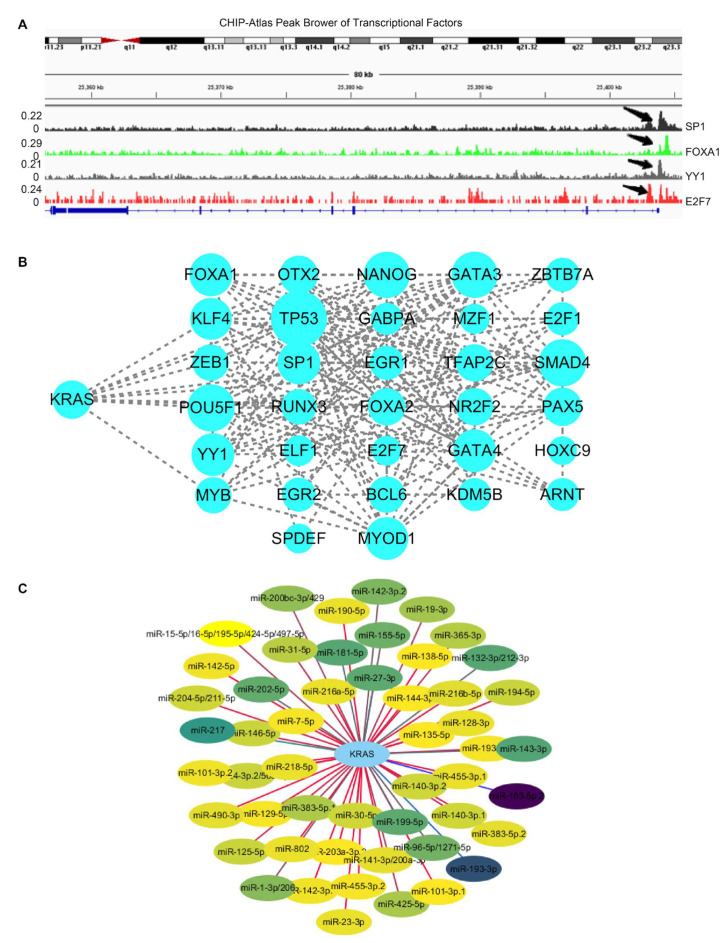
Regulators of KRAS. (**A**) IGV browser displaying the binding regions of transcriptional factors on the KRAS promoter regions. (**B**) PPI network showing the interaction among KRAS and transcriptional factors derived from ChIPBase V3.0. (**C**) Network indicating the interaction between KRAS and its target microRNAs derived from TargetScan 8.0.

**Figure 6 cells-11-01427-f006:**
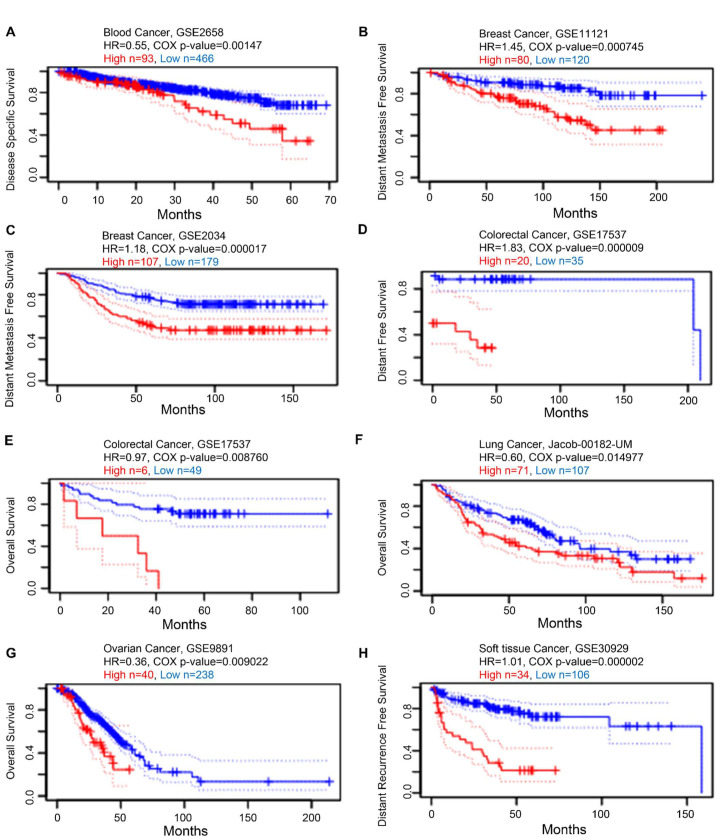
Kaplan–Meier survival curves comparing high and low expression in multiple cancer types from PrognoScan. (**A**) Disease-specific survival of blood cancer patients. (**B**,**C**) Distant metastasis-free survival of breast cancer patients. (**D**,**E**) Distant-free survival (**D**) and overall survival (**E**) of colorectal cancer patients. (**F**) Overall survival of lung cancer patients. (**G**) Overall survival of ovarian cancer patients. (**H**) Distant recurrence-free survival of soft tissue cancer patients. A *p*-value < 0.05 was considered a significant threshold.

**Figure 7 cells-11-01427-f007:**
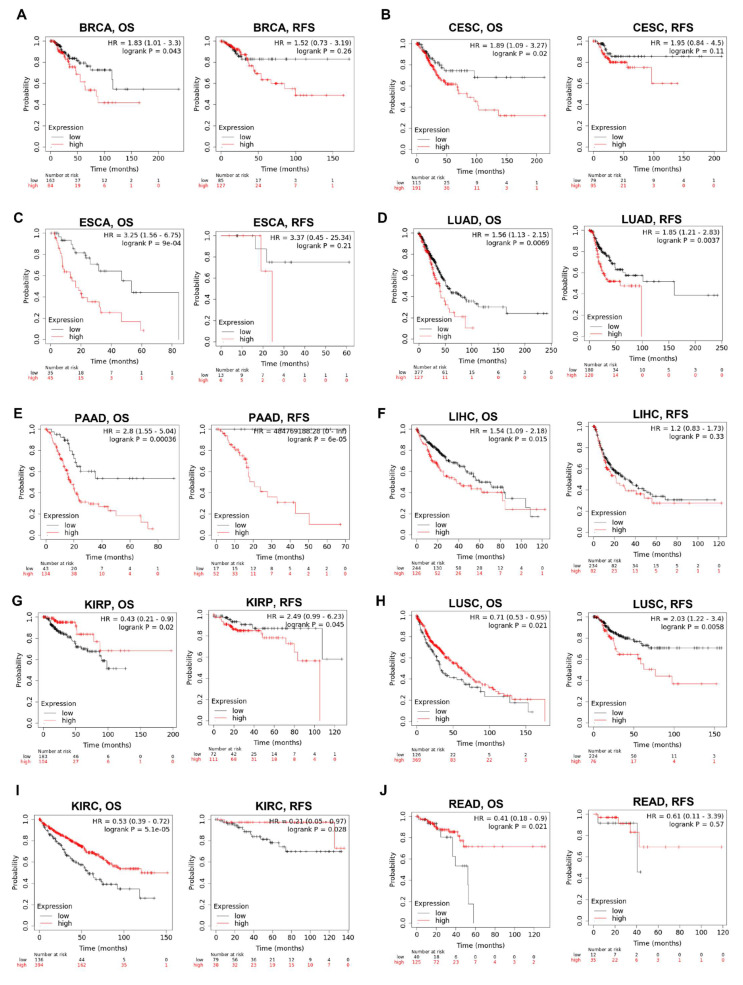
Kaplan–Meier survival curves comparing the high and low expression of KRAS in different cancer types. OS and RFS of BRCA (**A**), CESC (**B**), ESCA (**C**), LUAD (**D**), PAAD (**E**), LIHC (**F**), KIRP (**G**), LUSC (**H**), KIRC (**I**), and READ (**J**). OS, overall survival; RFS, relapse-free survival. A *p*-value < 0.05 was considered a significant threshold.

**Figure 8 cells-11-01427-f008:**
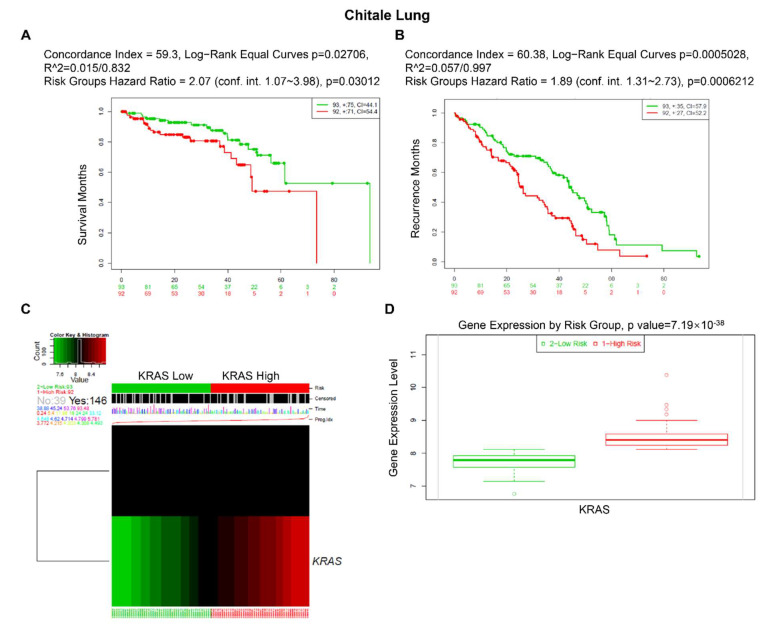
Expression level and prognostic values of KRAS for OS from the training cohorts of lung cancer via SurvExpress Platform. (**A**,**B**) Kaplan–Meier curves for survival (**A**) and recurrence (**B**) of KRAS between high- and low-risk groups. (**C**,**D**) Heatmap (**C**) and histogram figure (**D**) showing the KRAS level in high- and low-risk groups. A *p*-value < 0.05 was considered a significant threshold.

**Figure 9 cells-11-01427-f009:**
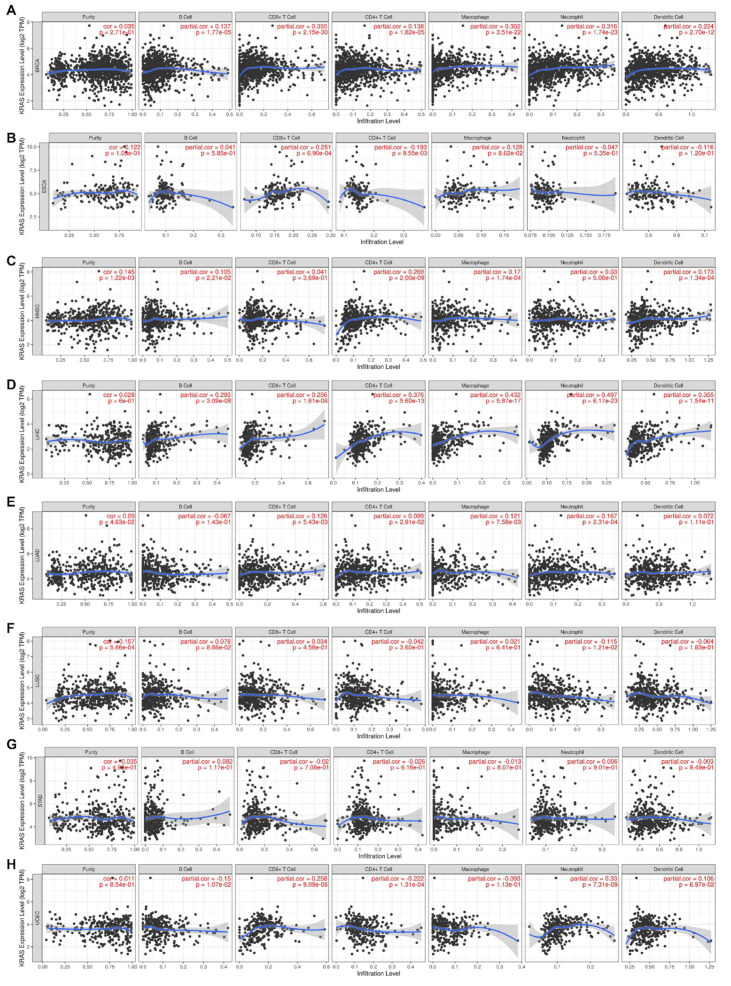
Correlation between KRAS and immune cell infiltration (B cells, CD8+T cells, CD4+T cells, macrophages, neutrophils, and dendric cells) in BRCA (**A**), ESCA (**B**), HNSC (**C**), LIHC (**D**), LUAD (**E**), LUSC (**F**), STAD (**G**), and UCES (**H**). A *p*-value < 0.05 was considered a significant threshold.

**Figure 10 cells-11-01427-f010:**
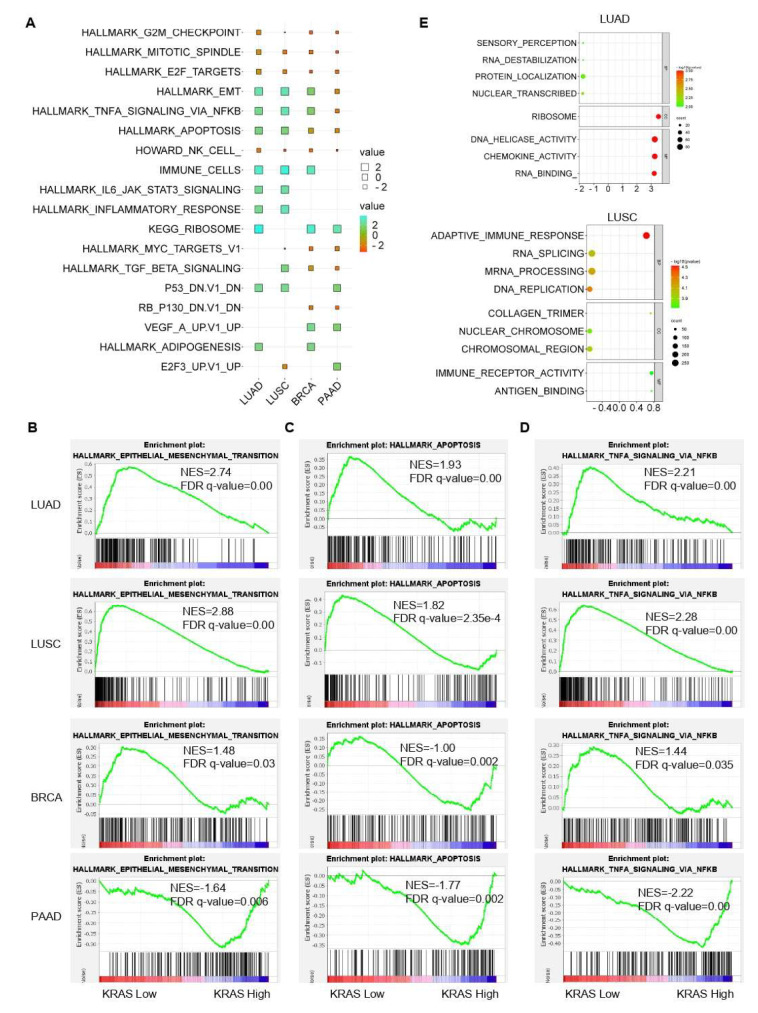
Gene signature analysis of KRAS-Low vs. KRAS-High datasets in LUAD, LUSC, BRCA, and PAAD. (**A**) Common gene signatures among different cancer types. (**B**–**D**), EMT (**B**) apoptosis (**C**) and TNFα/NFκB signaling (**D**) are enriched in LUAD, LUSC, BRCA, and PAAD. E, Bubble plots showing the Gene Ontology of KRAS-low vs KRAS-High datasets in LUAD (top) and LUSC (bottom). A FDR *q*-value < 0.05 was considered a significant threshold.

**Figure 11 cells-11-01427-f011:**
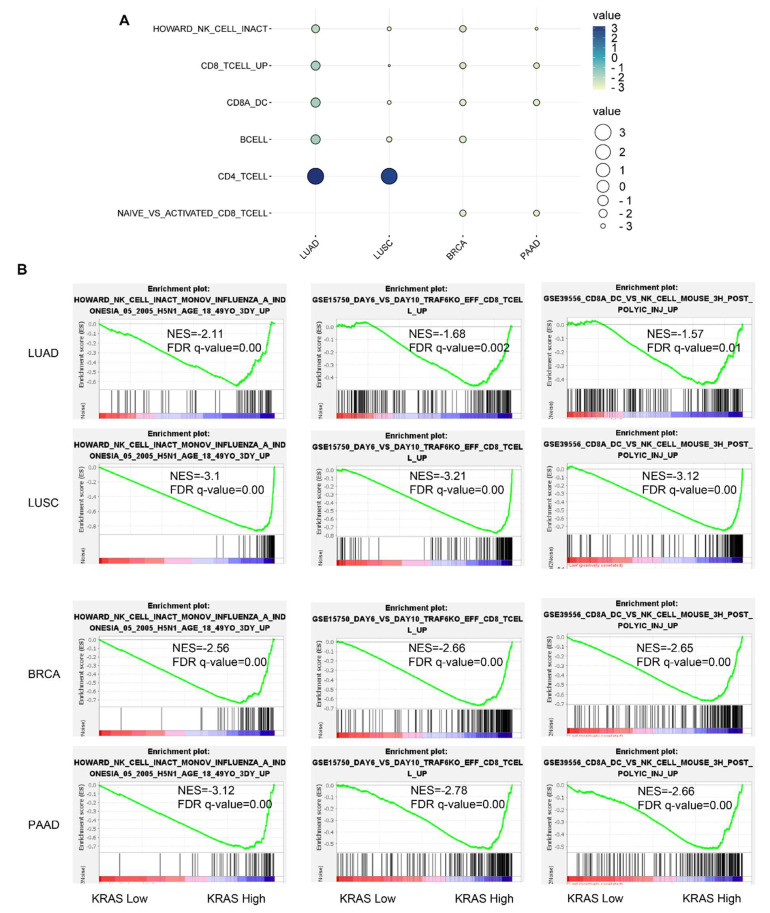
Immunological signature gene set analysis of KRAS-low vs. KRAS-high datasets in LUAD, LUSC, BRCA, and PAAD. (**A**) Bubble plot showing common gene signatures of immune cell infiltration among LUAD, LUSC, BRCA, and PAAD. (**B**) NK cell, CD8+T cell, and CD8A_Dentritic cell_NK cell infiltration signatures are enriched in LUAD, LUSC, BRCA, and PAAD. A FDR *q*-value < 0.05 was considered a significant threshold.

**Figure 12 cells-11-01427-f012:**
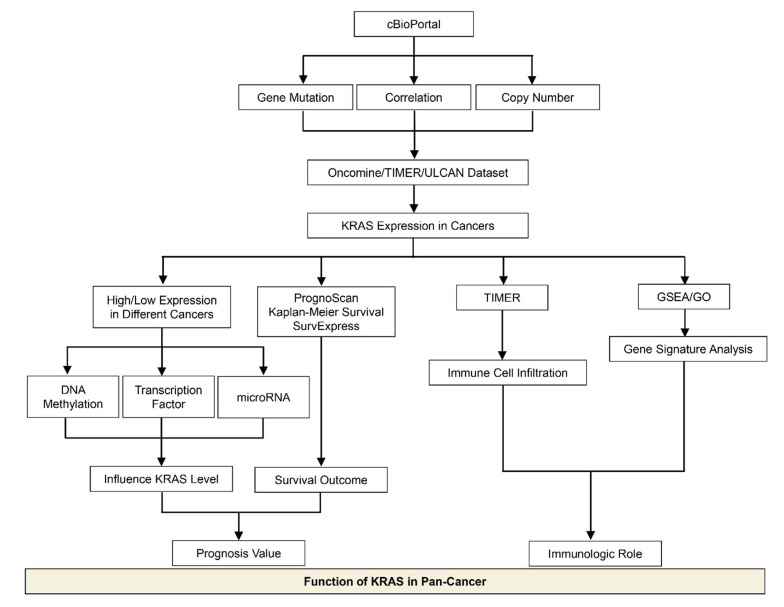
Schematic depicting the model of this study.

**Table 1 cells-11-01427-t001:** The top 20 frequently mutated genes in cancers.

Symbol	Affected Cases	CNV Gain	CNV Loss	Mutations
TP53	4796/12,538 (38.25%)	144/10,473 (1.37%)	495/10,473 (4.73%)	1423
PIK3CA	1621/12,538 (12.93%)	1367/10,473 (13.05%)	201/10,473 (1.92%)	450
FAT4	1415/12,538 (11.29%)	213/10,473 (2.03%)	378/10,473 (3.61%)	2225
KRAS	1333/12,538 (10.63%)	518/10,473 (4.95%)	182/10,473 (1.74%)	182
KMT2D	1310/12,538 (10.45%)	283/10,473 (2.70%)	300/10,473 (2.86%)	1673
PTEN	1205/12,538 (9.61%)	203/10,473 (1.94%)	949/10,473 (9.06%)	893
KMT2C	1137/12,538 (9.07%)	500/10,473 (4.77%)	718/10,473 (6.86%)	1535
ARID1A	1083/12,538 (8.64%)	212/10,473 (2.02%)	1128/10,473 (10.77%)	1006
FAT1	1058/12,538 (8.44%)	304/10,473 (2.90%)	836/10,473 (7.98%)	1478
APC	1049/12,538 (8.37%)	268/10,473 (2.56%)	518/10,473 (4.95%)	1128
BRAF	924/12,538 (7.37%)	539/10,473 (5.15%)	501/10,473 (4.78%)	235
NF1	913/12,538 (7.28%)	418/10,473 (3.99%)	450/10,473 (4.30%)	1110
ATRX	881/12,538 (7.03%)	184/10,473 (1.76%)	167/10,473 (1.59%)	1143
RNF213	881/12,538 (7.03%)	1047/10,473 (10.00%)	206/10,473 (1.97%)	1299
ZFHX3	833/12,538 (6.64%)	232/10,473 (2.22%)	430/10,473 (4.11%)	1114
GRIN2A	799/12,538 (6.37%)	355/10,473 (3.39%)	159/10,473 (1.52%)	906
TRRAP	763/12,538 (6.09%)	475/10,473 (4.54%)	177/10,473 (1.69%)	1020
ATM	732/12,538 (5.84%)	247/10,473 (2.36%)	731/10,473 (6.98%)	891
ERBB4	709/12,538 (5.65%)	338/10,473 (3.23%)	795/10,473 (7.59%)	836
AKAP9	709/12,538 (5.65%)	473/10,473 (4.52%)	160/10,473 (1.53%)	957

**Abbreviations:** CNV, copy number variation; TP53, tumor protein p53; PIK3CA, phosphatidylinositol-4,5-bisphosphate 3-kinase, catalytic subunit alpha; FAT4, FAT atypical cadherin 4; KRAS, Kirsten rat sarcoma viral oncogene homolog; KMT2D, lysine (K)-specific methyltransferase 2D; PTEN, phosphatase and tensin homolog; KMT2C, lysine (K)-specific methyltransferase 2C; ARID1A, AT rich interactive domain 1A (SWI-like); FAT1, FAT atypical cadherin 1; APC, adenomatous polyposis coli; BRAF, B-Raf proto-oncogene, serine/threonine kinase; NF1, neurofibromin 1; ATRX, alpha thalassemia/mental retardation syndrome X-linked; RNF213, ring finger protein 213; ZFHX3, zinc finger homeobox 3; GRIN2A, glutamate receptor, ionotropic, N-methyl D-aspartate 2A; TRRAP, transformation/transcription domain-associated protein; ATM, ATM serine/threonine kinase; ERBB4, erb-b2 receptor tyrosine kinase 4; AKAP9, A kinase (PRKA) anchor protein 9.

**Table 2 cells-11-01427-t002:** KRAS expression in different cancer patients.

Cancer	Cancer Number	Normal Number	Cancer Expression	Normal Expression	Fold Change	*p*-Value	FDR
BLCA	411	19	7.9	6.67	1.18	0.63	0.83
BRCA	1104	113	9.25	5.86	1.58	2.00 × 10^−18^	1.60 × 10^−17^
CHOL	36	9	6.39	2.5	2.55	4.20 × 10^−9^	4.20 × 10^−8^
COAD	471	41	8.49	11.55	0.73	1.00 × 10^−7^	6.70 × 10^−7^
ESCA	162	11	21.36	10.35	2.06	0.23	0.52
HNSC	502	44	7.93	6.61	1.2	0.049	0.11
KICH	65	24	7.71	6.74	1.14	0.26	0.43
KIRC	535	72	6.45	7.33	0.88	0.00021	0.00051
KIRP	289	32	6.58	5.73	1.15	0.42	0.63
LIHC	374	50	3.31	2.5	1.32	0.029	0.065
LUAD	526	59	11.56	6.59	1.75	3.10 × 10^−9^	1.90 × 10^−8^
LUSC	501	49	11.27	6.43	1.75	2.10 × 10^−11^	1.10 × 10^−10^
PAAD	178	4	8.17	6.62	1.23	0.49	0.94
PRAD	499	52	5.45	5.25	1.04	0.85	0.91
STAD	375	32	16.21	6.6	2.46	0.0003	0.0011
THCA	510	58	4.42	4.74	0.93	0.023	0.052
UCEC	548	35	6.59	4.73	1.4	0.034	0.09

**Table 3 cells-11-01427-t003:** Annotation information of DNA methylation sites.

Tumor	Probe ID	Average of Tumor Samples	Average of Normal Samples	*p*-Value
LUAD	cg20836156	0.619649	0.666993	1.89740 × 10^−9^
PAAD	cg12990174	0.768317	0.843160	0.00126474
	cg20836156	0.633778	0.707345	0.004233613
READ	cg17197538	0.051156	0.040148	9.32 × 10^−7^
COAD	cg01269191	0.056962	0.047081	9.95 × 10^−5^

**Table 4 cells-11-01427-t004:** Correlation of KRAS with BRCA and LUAD OS with different clinicopathological features.

Clinicopathological Features	BRCA OS			LUAD OS		
	N	Hazard Ratio	*p*	N	Hazard Ratio	*p*
**Stage**						
Stage 1	180	0.45 (0.16–1.2)	0.1	270	1.78 (1.04–3.03)	0.033
Stage 2	619	1.76 (1.08–2.86)	0.02	119	1.92 (1.09–3.35)	0.021
Stage 3	247	1.83 (1.01–3.3)	0.043	81	1.51 (0.82–2.79)	0.18
Stage 4	20	0 (0–inf)	7.8 × 10^−7^	26	2.93 (0.66–13.14)	0.14
**Mutation Burden**						
High	493	1.9 (1.18–3.07)	0.0071	255	1.72 (1.12–2.64)	0.012
Low	485	1.54 (0.93–2.54)	0.093	244	1.54 (0.97–2.44)	0.067
**Gender**						
Male	---	---	---	234	2.26 (1.48–3.46)	0.00011
Female	1077	1.48 (1.07–2.04)	0.016	270	1.23 (0.79–1.92)	0.36
**Race**						
White	752	1.76 (1.2–2.57)	0.0032	387	1.53 (1.08–2.15)	0.015
Asian	61	7204224608 (0–inf)	0.0054	---	---	---
Black of African American	181	0.4 (0.17–0.94)	0.03	52	1.96 (0.61–6.32)	0.25

**Table 5 cells-11-01427-t005:** Drug–gene interactions or potentially available drug categories with abnormal expression of KRAS.

Compound	Interaction Types	PMID	Interactions Score
AZD-4785	n/a	28615361	1.42
Panitumumab	n/a	20978259, 21398618, 18316791	1.31
Cetuximab	n/a	20978259, 28632865	0.84
Pelareorep	Inhibitor	24798549, 26156229	0.63
PD-0325901	n/a	21325073, 20570890, 26582713	0.52
XMT-1536	n/a	-	0.47
Chembl217354	n/a	25665005	0.47
Rilotumumab	n/a	24919569	0.47
Ralimetnib	n/a	26725216	0.47
SAR-125844	n/a	25504634	0.47
Necitumumab	n/a	26766738	0.47
Imgatuzumab	n/a	23209031	0.47
Selumetinib	Inhibitor	25870145, 27312529, 27556948	0.43
Ridaforolimus	n/a	26725216	0.32
Phenformin	n/a	26574479	0.32
Teprotumumab	n/a	21985784	0.32

## Data Availability

The datasets used to support the results of this study are available from the corresponding author upon request.

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
