# Peer review of "Prognostic and Immunotherapeutic Roles of KRAS in Pan-Cancer"

_cells, 2022, doi:10.3390/cells11091427_

Round 1
Reviewer 1 Report
In this article, the authors have highlighted the importance of KRAS and its link to the immune cell infiltration of different tumor entities. Authors have analyzed KRAS mutations in different tumor entities and showed that KRAS mutated and non-mutated patients do not differ in terms of overall survival. KRAS expression was higher in Gain and Amplification cohorts compared to other genetic alterations as expected. Authors have examined the prognostic value of KRAS in different cancer types and depicted the association between high KRAS levels and poor prognosis of several cancers. Additionally, they have shown the correlation between KRAS and immune cell infiltrates in different cancers. Lastly, they finalize their findings with gene signature analyses in LUAD, LUSC, BRCA, and PAAD. Regarding different platforms and detailed analyses, this article holds great importance. With different aspects, the field would use it as a source, because there are a lot of open-end questions in this manuscript. Identification of prognostic and IT roles of KRAS in a tumor agnostic manner should be supported by several aspects.
First of all, the first 5 figures are within the line of the manuscript. After these figures, the authors have analyzed the correlation between KRAS and immune cell infiltrates. However, the previous figures and 6. figure are detached from each other. If authors want to highlight the prognostic value of KRAS, they should have followed up the correlation between KRAS levels and the immune cell profile of patients from different cancer types more systematically. For example, they could have revealed the link between gene signature enrichments and immune cell profiles in patients harboring low and high KRAS. However, instead of checking this link, additional results are showing the methylation status of KRAS in tumors and normal tissues. I found it confusing. If they want to mention more about the prognostic meaning of KRAS, they should have done different analyses showing chemotherapy response, and metastatic status of patients. However, I have seen nothing regarding these points. And if they want to reveal methylation status and DNA methylation, they could have included this dimension in the title. For this reason, this manuscript should be streamlined. Otherwise, it seems like mining of different platforms considering KRAS. In the field, we all know that KRAS is an important oncogenic driver and of course, it will have an influence on survival in different tumors. Researchers should put more data regarding their title and the line of this manuscript. As they have indicated in the discussion, they can also show how different mutations of KRAS could influence patient survival and other prognostic outcomes. Maybe, they could also provide immune cell profiles for different specific KRAS mutations (like KRASG12D, KRASG12C) if it is possible. I believe with these points, the manuscript will remain in its descriptive form but it will provide more insights regarding the title and it will be more useful for the field.
Reviewer 2 Report
The authors present a manuscript based on a wide range of meta analyses on data form publicly available sources – all of which aiming to draw a comprehensive picture of the usability of KRAS as prognostic marker in pan-cancer.
The study is well-designed and covers a large number of different aspects of KRAS biology – a very comprehensive analysis indeed.
The data are presented in a logic and straightforward manner with each figure and table being carefully assembled and presented.
The authors conclude with a short, yet good discussion with clear take home messages. They also mention potential shortcomings in their data analysis. The manuscript is well written and manages to present the large amount of data in a very accessible way.
Overall, a very useful and very comprehensive addition to the current literature on KRAS.
I have only a few suggestions:
-Even though the expression “pan-cancer” is rather clear, the authors sometime use this word seemingly as if “pan-cancer” would represent its own entity of cancers. For example: line 17: The study of KRAS in pan-cancer remains sparse. Even though it’s clear what is intended, this sentence could be changed to “comparative studies of the relevance of KRAS across diverse tumors remains sparse”. The authors should also check the other positions within the text where “pan-cancer” is mentioned.
-The title could be adjusted to clearly reflect the nature of this manuscript: xxxxx -a meta analysis across pan-cancer data.
- Even though information about statistical methods are provided in the methods section, it would be good to include any such information also in all figure legends.
- Sentences in lines in line 17, 20, 31, 84, 95, 98,295, 355, 368/369, 382, 393,496 should be checked again.
-line 142: the word “censored” seems to be not right. Please explain or change
-line 423: In this review … should be changed to : In this meta analysis
-Figure 10 A: shape of arrows should be changed to clearer ones.
-Supplementary Tables 1 and 2 were not provided in the Supplemental file
-Figure legends for supplementary figures should be expanded with more information.
Lastly: Personally, I think a general overview figure of the mode of action of KRAS in the Introduction would enhance this manuscript. Also, some summary graphs of mutations on KRAS would allow easier access to the produced data.
Round 2
Reviewer 1 Report
The authors have made significant changes in the flow of data presentation. I think this revised version is totally worth to be published. I am pretty sure, the field will benefit from this paper a lot. Congratulations to the authors!